# Med-cDiff: Conditional Medical Image Generation with Diffusion Models

**DOI:** 10.3390/bioengineering10111258

**Published:** 2023-10-28

**Authors:** Alex Ling Yu Hung, Kai Zhao, Haoxin Zheng, Ran Yan, Steven S. Raman, Demetri Terzopoulos, Kyunghyun Sung

**Affiliations:** 1Computer Science Department, University of California, Los Angeles, CA 90095, USA; haoxinzheng@g.ucla.edu (H.Z.); dt@cs.ucla.edu (D.T.); 2Department of Radiology, University of California, Los Angeles, CA 90095, USA; kz@kaizhao.net (K.Z.); ranyan@mednet.ucla.edu (R.Y.); sraman@mednet.ucla.edu (S.S.R.); ksung@mednet.ucla.edu (K.S.); 3Bioengineering Department, University of California, Los Angeles, CA 90095, USA; 4VoxelCloud, Inc., Los Angeles, CA 90024, USA

**Keywords:** image generation, diffusion models, generative models, super-resolution, denoising, inpainting

## Abstract

Conditional image generation plays a vital role in medical image analysis as it is effective in tasks such as super-resolution, denoising, and inpainting, among others. Diffusion models have been shown to perform at a state-of-the-art level in natural image generation, but they have not been thoroughly studied in medical image generation with specific conditions. Moreover, current medical image generation models have their own problems, limiting their usage in various medical image generation tasks. In this paper, we introduce the use of conditional Denoising Diffusion Probabilistic Models (cDDPMs) for medical image generation, which achieve state-of-the-art performance on several medical image generation tasks.

## 1. Introduction

Conditional image generation refers to the generation of images using a generative model based on relevant information, which we denote as a condition. When the condition is an image, this is also referred to as image-to-image translation. In the medical domain, this has many important applications such as super-resolution, inpainting, denoising, etc., which can potentially improve healthcare [1]. Super-resolution can help shorten imaging time and improve imaging quality. Denoising helps clinicians and downstream algorithms to make better diagnostic judgments. Medical image inpainting can be beneficial to anomaly detection.

Existing generative models are able to perform some of these jobs decently; e.g., the Hierarchical Probabilistic UNet (HPUNet) [2] for ultrasound image inpainting, and the progressive Generative Adversarial Network (GAN) [3] and SMORE [4] for medical image super-resolution. These methods work to some extent, but they are tailored to specific applications or imaging modalities, making it difficult for researchers to adapt them to different tasks or modalities. MedGAN [5] and UP-GAN [6] target general-purpose medical image generation; however, they are too challenging to train and/or produce underwhelming results.

Models based on Variational Autoencoders (VAE) can be effective in some medical applications [2,7], but the generated images tend to be blurry [8]. Although GAN-based models can generate high-quality medical images [5,9], they suffer from unstable training due to vanishing gradient, convergence, and mode collapse [10]. Normalizing Flow (NF), which has also been used in medical imaging [11,12], can estimate the exact likelihood of the generated sample, making it suitable for certain applications; however, NF requires specifically designed network architectures and the generated image quality fails to impress.

Diffusion models have been dominant in natural image generation due to their ability to generate high-fidelity realistic images [13,14,15,16]. They have also been applied to medical image generation [17,18,19,20], such as in super-resolution medical imaging [21], but there are only a limited number of studies using conditional diffusion models.

We propose a conditional Denoising Diffusion Probabilistic Model (cDDPM), which we call the medical conditional diffusion model (Med-cDiff), and apply it to a variety of medical image generation tasks, including super-resolution, denoising, and inpainting. In a series of experiments, we show that Med-cDiff achieves state-of-the-art (SOTA) generation performance on these tasks, which demonstrates the great potential of diffusion models in conditional medical image generation.

## 2. Related Work

Before diffusion models became popular in medical image analysis or in mainstream computer vision, GANs [22] were the most popular image generation methods. Developed to perform conditional natural image generation, Pix2PixGAN [23] was adapted to medical imaging and several researchers have shown its usefulness in such tasks [24,25,26,27]. Zhu et al. [28] proposed CycleGAN to perform conditional image-to-image translation between two domains using unpaired images, and the model has also been extensively used in medical imaging. Du et al. [29] made use of CycleGAN in CT image artifact reduction. Yang et al. [30] used a structure-constrained CycleGAN to perform unpaired MRI-to-CT brain image generation. Liu et al. [31] utilized multi-cycle GAN to synthesize CT images from MRI for head-neck radiotherapy. Harms et al. [32] applied CycleGAN to image correction for cone-beam computed tomography (CBCT). Karras et al. [33] proposed StyleGAN, which has an automatically learned, unsupervised separation of high-level attributes and stochastic variation in the generated images, enabling easier control of the image synthesis process. Fetty et al. [34] manipulated the latent space for high-resolution medical image synthesis via StyleGAN. Su et al. [35] performed data augmentation for brain CT motion artifacts detection using StyleGAN. Hong et al. [9] introduced 3D StyleGAN for volumetric medical image generation. Other GAN-based methods have also been proposed for medical imaging. Progressive GAN [3] was used to perform medical image super-resolution. Upadhyay et al. [6] extended the model by utilizing uncertainty estimation to focus more on the uncertain regions during image generation. Armanious et al. [5] proposed MedGAN, specific to medical image domain adaptation, which captured the high and low frequency components of the desired target modality.

Apart from GANs, other generative models, including VAEs and NFs, are also popular in image generation. The VAE was introduced by Kingma and Welling [36], and it has been the basis for a variety of methods for image generation. Vahdat and Kautz [37] developed Nouveau VAE (NVAE), a hierarchical VAE that is able to generate highly realistic images. Hung et al. [2] adapted some of the features from NVAE into their hierarchical conditional VAE for ultrasound image inpainting. Cui et al. [38] adopted NVAE in positron emission tomography (PET) scan image denoising and uncertainty estimation. As for the NF models, Grover et al. [39] proposed AlignFlow based on a similar concept with NF models instead of GANs. Bui et al. [40] extended AlignFlow into medical imaging for Unpaired multi-contrast MRI conditonal image generation. Wang et al. [41] and Beizaee et al. [42] applied NF to medical image harmonization.

In recent years, diffusion models have become the most dominant algorithm in image generation due to their ability to generate realistic images. On natural images, diffusion models have achieved SOTA results in unconditional image generation by outperforming their GAN counterparts [13,14]. Diffusion models have achieved outstanding performance in tasks such as super-resolution [16,43], image editing [44,45], and unpaired conditional image generation [46], and they have attained SOTA performance in conditional image generation [15]. In medical imaging, unsupervised anomaly detection is an important application of unconditional diffusion models [17,47,48,49]. Image segmentation is a popular application of conditional diffusion models, where the image to be segmented is used as the condition [19,50,51,52,53]. Diffusion models have also been widely applied to accelerating MRI reconstruction [20,54,55]. Özbey et al. [18] used GANs to shorten the denoising process in diffusion models for medical imaging.

## 3. Methods

### 3.1. Background

The goal of conditional image generation is to generate the target image $x_{0}$ given a correlated conditional image *y*. Diffusion models consist of two parts: a forward noising process *q*, and a reverse denoising process $p_{\theta }$ parameterized by θ. Figure 1 illustrates conditional diffusion models. At a high level, given *y*, they sample from a data distribution during $p_{\theta }$, reversing *q*, which adds noise iteratively to the original image $x_{0}$. More specifically, the sampling process starts with a random noise sample $x_{T}$, and iteratively generates less-noisy samples, $x_{T-1},x_{T-2},\cdots$, based on the conditional image *y* for *T* steps until reaching the final output sample $x_{0}$. For a specific sample $x_{t}$ during the process, the larger *t* is, the more noisy the sample will be. Given the conditional image *y*, the reverse process $p_{\theta }$ learns to denoise the sample $x_{t}$ by one step to $x_{t-1}$.

The forward process *q* is a Markovian noising process, where Gaussian noise is added to the image $x_{t-1}$ at each time step t=1,2,…,T according to a variance schedule $\beta_{t}$:(1)$$q\left(x_{t}|x_{t-1}\right)=N\left(x_{t};\sqrt{1-\beta_{t}}x_{t-1},\beta_{t}I\right),$$
where N(·) denotes the normal distribution and *I* is the identity matrix. Note that
(2)$$q\left(x_{1},\ldots ,x_{T}|x_{0}\right)=\prod_{t=1}^{T}q\left(x_{t}|x_{t-1}\right),$$
where *T* is the number of steps. The forward noising process (Equation 1) can be used to sample $x_{t}$ at any timestep *t* in closed form. In other words, since
(3)$$q\left(x_{t}|x_{0}\right)=N\left(x_{t};\sqrt{\bar{\alpha_{t}}}x_{0},\left(1-\bar{\alpha_{t}}\right)I\right),$$
then for the original image $x_{0}$ and any given timestep *t*
(4)$$x_{t}=\sqrt{\bar{\alpha_{t}}}x_{0}+\left(1-\bar{\alpha_{t}}\right)\epsilon ,$$
where $\alpha_{t}=1-\beta_{t}$ and $\bar{\alpha_{t}}=\prod_{i=1}^{t}\alpha_{i}$, and ϵ∼N(0,1). When *T* is large, we can assume that $x_{T}∼N\left(0,I\right)$, which is random Gaussian noise containing no information regarding the original image $x_{0}$ [13].

In a conditional diffusion model, the objective is to learn the reverse process $p_{\theta }$ so that we can infer $x_{t-1}$ given $x_{t}$ and the conditional image *y*. In this way, starting from the Gaussian noise $x_{T}∼N\left(0,1\right)$, and given *y*, we can iteratively infer the sample at time step t−1 from the sample at time step *t* until we reach the original image $x_{0}$. For the reverse process,
(5)$$p_{\theta }\left(x_{0},\ldots ,x_{T}|y\right)=p_{\theta }\left(x_{T}\right)\prod_{t=1}^{T}p_{\theta }\left(x_{t-1}|x_{t},y\right).$$

The reverse process can therefore be parameterized as
(6)$$p_{\theta }\left(x_{t-1}|x_{t},y\right)=N\left(x_{t-1};\mu_{\theta }\left(x_{t},y,t\right),\Sigma_{\theta }\left(x_{t},y,t\right)\right),$$
where we set $\Sigma_{\theta }\left(x_{t},y,t\right)=\sigma_{t}^{2}I$. As for $\mu_{\theta }\left(x_{t},y,t\right)$, Ho et al. [13] showed that it must be parameterized as
(7)$$\mu_{\theta }\left(x_{t},y,t\right)=\frac{1}{\sqrt{\alpha_{t}}}\left(x_{t}-\frac{\beta_{t}}{\sqrt{1-\bar{\alpha_{t}}}}\epsilon_{\theta }\left(x_{t},y,t\right)\right),$$
where $\epsilon_{\theta }\left(x_{t},y,t\right)$ is a function approximating ϵ.

For a total of *T* steps, the training objective is to minimize the variational lower bound on the negative log-likelihood:(8)$$\begin{array}{cc} E\left[-\log p_{\theta }\left(x_{0}|y\right)\right] & \leq E_{q}\left[-\log\frac{p_{\theta }\left(x_{0},\ldots ,x_{T}|y\right)}{q(x_{1},\ldots ,x_{T}|x_{0})}\right]\\ \; & =E_{q}\left[-\log p_{\theta }\left(x_{T}\right)-\sum_{t=1}^{T}\log\frac{p_{\theta }\left(x_{t-1}|x_{t},y\right)}{q(x_{t}|x_{t-1})}\right]\\ & =L(\theta ). \end{array}$$

More efficient training can be achieved by optimizing random terms in the training objective L(θ) using stochastic gradient descent. Therefore, we can rewrite the training objective as
(9)$$L\left(\theta \right)=E_{q}\left[\sum_{t=1}^{T}L_{t}\left(\theta \right)\right],$$
where
(10)$$L_{t}\left(\theta \right)=\left\{\begin{array}{cc} -\log p_{\theta }\left(x_{0}|x_{1}\right) & \mathrm{if}\;t=0,\\ D_{KL}(q\left(x_{t}|x_{t+1},x_{0}\right)∥p_{\theta }\left(x_{t}|x_{t+1},y\right)) & \mathrm{if}\;0<t<T,\\ D_{KL}(q\left(x_{T}|x_{0}\right)∥p_{\theta }\left(x_{T}\right)) & \mathrm{if}\;t=T, \end{array}\right.$$
and $D_{KL}(.∥.)$ is the Kullback-Leibler (KL) divergence between two distributions. In (Equation 10), the term $q(x_{t}|x_{t+1},x_{0})$ is given by
(11)$$q\left(x_{t}|x_{t+1},x_{0}\right)=N\left(x_{t};{\tilde{\mu }}_{t+1}\left(x_{t+1},x_{0}\right),{\tilde{\beta }}_{t+1}I\right),$$
where
(12)$${\tilde{\mu }}_{t+1}\left(x_{t+1},x_{0}\right)=\frac{\sqrt{{\bar{\alpha }}_{t}}\beta_{t+1}}{1-{\bar{\alpha }}_{t+1}}x_{0}+\frac{\sqrt{\alpha_{t+1}}\left(1-{\bar{\alpha }}_{t}\right)}{1-{\bar{\alpha }}_{t+1}}x_{t+1},$$
with
(13)$${\tilde{\beta }}_{t+1}=\frac{1-{\bar{\alpha }}_{t}}{1-{\bar{\alpha }}_{t+1}}\beta_{t+1}.$$

### 3.2. Training and Sampling

When t=T, $L_{T}\left(\theta \right)$ is a constant with no learnable parameters since $\beta_{t}$ is fixed to a constant. Therefore, $L_{t}\left(\theta \right)$ can be ignored during training.

When 0<t<T, $L_{t}\left(\theta \right)$ can be expressed as
$$\begin{array}{ccc} L_{t}\left(\theta \right) & =E_{q}\left[\frac{{∥{\tilde{\mu }}_{t+1}\left(x_{t+1},x_{0}\right)-\mu_{\theta }\left(x_{t+1},y,t+1\right)∥}^{2}}{2\sigma_{t+1}^{2}}\right]+C & \left(14\right)\\ & =E_{x_{0},\epsilon }\left[\frac{{∥\frac{1}{\sqrt{{\bar{\alpha }}_{t+1}}}\left(x_{t+1}\left(x_{0},\epsilon \right)-\frac{\beta_{t+1}}{\sqrt{1-{\bar{\alpha }}_{t+1}}}\epsilon \right)-\mu_{\theta }\left(x_{t+1}\left(x_{0},\epsilon \right),y,t+1\right)∥}^{2}}{2\sigma_{t+1}^{2}}\right]+C & \left(15\right)\\ & =E_{x_{0},\epsilon }\left[\frac{\beta_{t+1}^{2}{∥\epsilon -\epsilon_{\theta }\left(\sqrt{{\bar{\alpha }}_{t+1}}x_{0}+\sqrt{1-{\bar{\alpha }}_{t+1}}\epsilon ,y,t+1\right)∥}^{2}}{2\sigma_{t+1}^{2}\alpha_{t+1}\left(1-{\bar{\alpha }}_{t+1}\right)}\right]+C, & \left(16\right) \end{array}$$
where *C* is a constant.

When t=0, assuming all the image data have been re-scaled to [−1,1], the expression of $L_{0}\left(\theta \right)$ can be written as
(17)$$L_{0}\left(\theta \right)=-\log p_{\theta }\left(x_{0}|x_{1}\right)=-\sum_{i=1}^{H}\sum_{j=1}^{W}\int_{f(x_{0}^{i,j}-\delta )}^{f(x_{0}^{i,j}+\delta )}N\left(x;\mu^{i,j},\theta \left(x_{1},1\right),\sigma_{1}^{2}\right)\;dx,$$
where *H* and *W* are the height and width of the image, respectively, and δ is a small number, and where
(18)$$f\left(x\right)=\left\{\begin{array}{cc} 1 & \mathrm{if}\;x>1,\\ x & \mathrm{if}\;-1<x<1,\\ -1 & \mathrm{if}\;x<-1. \end{array}\right.$$

From Equations (14) and (17), we see that the training objective is differentiable with respect to the model parameter θ. During each training step, we sample the image pair $\left(x_{0},y\right)$ from the dataset $x_{0}$, $y∼p_{data}\left(x,y\right)$, the time step *t* from a uniform distribution t∼U({1,2,…,T}), and ϵ from a normal distribution ϵ∼N(0,I). We then perform gradient descent on
(19)$$\nabla_{\theta }{∥\epsilon -\epsilon_{\theta }\left(\sqrt{\bar{\alpha_{t}}}x_{0}+\sqrt{1-\bar{\alpha_{t}}}\epsilon ,y,t\right)∥}^{2},$$
which is an alternative variational lower bound that has been shown to be better for sampling quality [13].

During sampling, $x_{T}$ is first sampled from a normal distribution $x_{T}∼N\left(0,I\right)$. Then we iteratively sample $x_{T-1},x_{T-2},\cdots ,x_{0}$ from distribution $x_{t-1}∼p_{\theta }\left(x_{t-1}|x_{t},y\right)$ by
(20)$$x_{t-1}=\frac{1}{\sqrt{\alpha_{t}}}\left(x_{t}-\frac{\beta_{t}}{\sqrt{1-\bar{\alpha_{t}}}}\epsilon_{\theta }\left(x_{t},y,t\right)\right)+\sigma_{t}z,$$
where $\sigma_{t}$ is an untrained time dependent constant and z∼N(0,I).

## 4. Experiments

### 4.1. Datasets

Our method is evaluated on the following datasets:MRI Super Resolution: The dataset consists of 296 patients who underwent pre-operative prostate MRI prior to robotic-assisted laparoscopic prostatectomy. T2-weighted imaging was used for the experiment, acquired by the Turbo Spin Echo (TSE) MRI sequence following the standardized imaging protocol of the European Society of Urogenital Radiology (ESUR) PI-RADS guidelines [56]. Additionally, the dataset includes annotation of the transition zone (TZ) and peripheral zone (PZ) of the prostate. Overall, 238, 29, and 29 patients were used for training, validation, and testing, respectively. To perform super-resolution, we downsampled the images by a factor of $2\sqrt{2}$, 4, $4\sqrt{2}$, 8, $8\sqrt{2}$, and 16.X-ray Denoising: The public chest X-ray dataset [57] contains 5863 X-ray images with pneumonia and normal patients. Overall, 624 images were used for testing. Pneumonia patients were further categorized as virus- or bacteria-infected patients. We randomly added Gaussian noise as well as salt and pepper noise to the images and used the original images as the ground truth.MRI Inpainting: The dataset consists of 18,813 T1-weighted prostate MRI images that were acquired by the Spoiled Gradient Echo (SPGR) sequence. We used 6271 of them for testing. The masks were randomly generated during training, and they were fixed among different tests for testing.

### 4.2. Implementation and Evaluation Details

For Med-cDiff, $\epsilon_{\theta }\left(x_{t},y,t\right)$ was parameterized by a U-Net [58] while using group normalization [59]. The total number of steps was set to T=2000. The forward process variances were set to constants that linearly increase from $\beta_{1}=10^{-4}$ and $\beta_{T}=0.02$. We also set $\sigma_{t}^{2}=\beta_{t}$. All the images used were resized to 128×128, and the pixel values are normalized to the range [−1,1] in a patient-wise manner. The models were all trained for $2\times 10^{5}$ iterations with a learning rate of $1\times 10^{-4}$.

For quantitative evaluation, we used the following metrics: Learned Perceptual Image Patch Similarity (LISPS) (v1.0) [60] with AlexNet [61] as the backbone, Fréchet Inception Distance (FID) [62], accutance (acc) [63], which measures the sharpness of an image, Dice similarity coefficient (DSC) [64], classification accuracy, and the 2-alternative forced-choice (2AFC) paradigm [65].

Due to the domain gaps [66,67] between different datasets and different tasks, combining datasets and training a combined network would yield a worse performance than separately training the networks. Thus, we trained and tested our methods on different tasks separately.

### 4.3. MRI Super-Resolution

For MRI super-resolution, we downsampled the MRI images by a factor of $2\sqrt{2}$, 4, and $4\sqrt{2}$, and then we upscaled the images to their original size. We compared the performance of Med-cDiff against bilinear interpolation, pix2pixGAN [23], and SRGAN [68] both visually and quantitatively, evaluated by LPIPS, FID, and accutance, as well as performance comparison on the downstream zonal segmentation task.

Figure 2 shows qualitative results. Clearly, images generated by the other methods are blurry and lack realistic textures, whereas Med-cDiff is able to recover the shape of the prostate as well as relevant textures. For zonal segmentation, we utilized the pretrained CAT-nnUNet [69] and calculated the 3D patient-wise DSC for evaluation. The quantitative results are reported in Table 1, confirming that the images generated by Med-cDiff are the most realistic with the best sharpness and are useful in downstream zonal segmentation. Furthermore, to show the effectiveness of Med-cDiff on zonal segmentation, we further downsampled the original images by a factor of 8, $8\sqrt{2}$, and 16 and performed MRI super-resolution. The results on downstream zonal segmentation are plotted in Figure 3, which reveals that Med-cDiff clearly outperforms bilinear interpolation and pix2pixGAN. CAT-nnUNet performs similarly on images generated by Med-cDiff and SRGAN for PZ segmentation, but it performs better on images generated by Med-cDiff for TZ segmentation. The segmentation performance using bilinear interpolation and pix2pixGAN drops drastically as the upscaling factor increases, while the segmentation performance using images generated by SRGAN and Med-cDiff does not decrease much.

### 4.4. X-ray Denoising

We evaluated the denoising results using the LPIPS and FID metrics, and further evaluated the results by comparing the downstream classification performance, where 3-class classification (normal/bacterial pneumonia/viral pneumonia) was performed using VGG11 [70]. We compared Med-cDiff against pix2pixGAN [23] and UP-GAN [6].

The quantitative results are reported in Table 2. Med-cDiff outperforms the other methods in every metric. Qualitative results are shown in Figure 4, where we see that pix2pixGAN creates new artifacts and distorts the anatomy while UP-GAN creates unrealistic blurry images lacking details. More specifically, in the normal image example in Figure 4, the yellow arrows point to the newly generated artifacts, and the red arrows point to the unusually large spinal cord. By contrast, Med-cDiff generates realistic patterns in those regions. In the viral pneumonia example, pix2pixGAN cannot generate the bright pattern in the original image at the yellow arrow. As for the bacterial pneumonia example, pix2pixGAN cannot generate the spinal cord with the correct shape at the yellow arrow. In both pneumonia examples, pix2pixGAN failed to recover the correct shape of the ribs at the red arrows.

### 4.5. MRI Inpainting

We compared our method against other inpainting methods such as pix2pixGAN, HPUNet [2], and UP-GAN using the LPIPS and FID metrics. Furthermore, we performed a 2AFC paradigm [65] to measure how well trainees can discriminate real images from the generated ones. We randomly sampled 50 real and generated image pairs from the test set for each method and asked four trainees to perform 2AFC. We averaged the results from the four trainees.

The quantitative results in Table 3 reveal that Med-cDiff can generate the most realistic images. The 2AFC values convey that it is difficult to determine that images generated by Med-cDiff are not real, while it is easy to discern the inauthenticity of images generated by competing methods. The visual results in Figure 5 further confirm that Med-cDiff generates the most authentic images. More specifically, in the masked regions, pix2pixGAN generates unrealistic patterns that are clear indicators of images generated by GANs, while HPUNet can generate somewhat realistic patterns, although the generated patches are still relatively blurry. HPUNet was designed for ultrasound image inpainting, but the performance is unimpressive when applied to MRI images. This shows the difficulties in applying some methods to cross-imaging modalities. As for UP-GAN, the generated patches were blurry, while Med-cDiff generated realistic patterns and contents.

## 5. Conclusions

We have introduced Med-cDiff, a conditional diffusion model for medical image generation, and shown that Med-cDiff is effective in several medical image generation tasks, including MRI super-resolution, X-ray image denoising, and MRI image inpainting. We have demonstrated that Med-cDiff can generate high-fidelity images, both quantitatively and qualitatively superior to those generated by other GAN- and VAE-based methods. The images generated by Med-cDiff were also tested in downstream tasks such as organ segmentation and disease classification, and we showed that these tasks can benefit from the images generated by Med-cDiff.

More importantly, Med-cDiff was not designed for any specific application yet it outperforms models designed for specific applications. For example, SRGAN is specifically designed to generate high-resolution images from low-resolution images as it upsamples the low-resolution images within the network, while HPUNet is mainly used for inpainting ultrasound images to generate realistic ultrasound noise patterns. By contrast, since conditional diffusion models can generate highly realistic images, Med-cDiff can learn to generate various medical images with different characteristics and patterns.

In future work, we will apply Med-cDiff to other downstream tasks; e.g., anomaly detection and faster image reconstruction. Conditional medical image generation is not limited to these tasks. Other applications, such as inter-modality image translation and image enhancement, are also worthy of exploration.

## Figures and Tables

**Figure 1 bioengineering-10-01258-f001:**
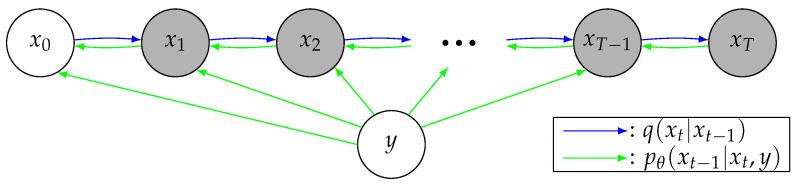
A graphical model representation of conditional diffusion models. The blue and green arrows indicate the forward and reverse processes, respectively.

**Figure 2 bioengineering-10-01258-f002:**
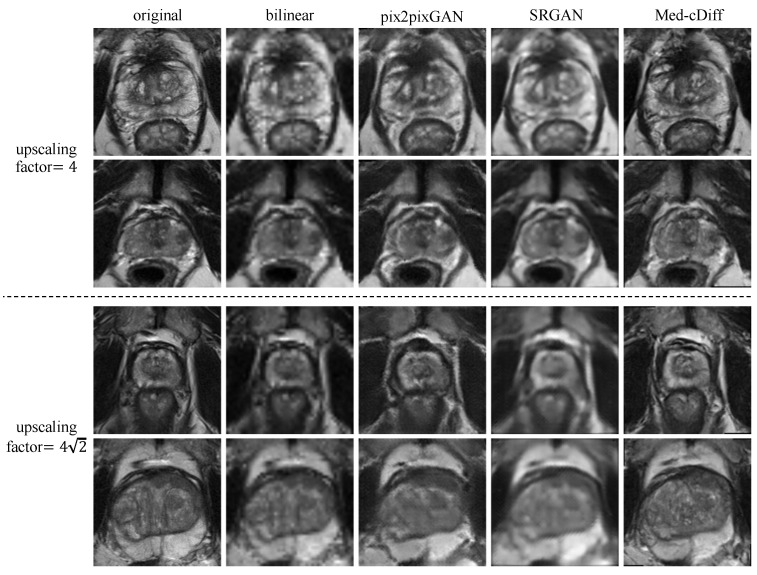
Qualitative comparison of Med-cDiff against other super-resolution methods.

**Figure 3 bioengineering-10-01258-f003:**
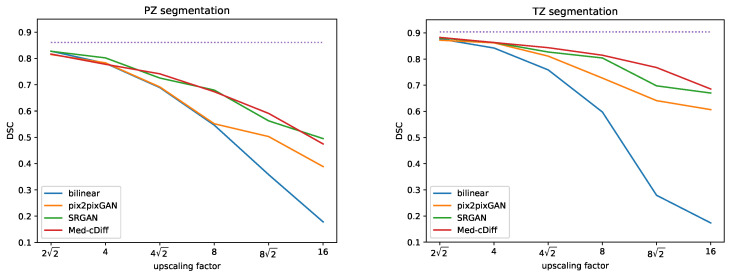
DSC comparison of Med-cDiff against bilinear interpolation, pix2pixGAN, and SRGAN for zonal segmentation. The purple dotted lines indicate scores from the original high-resolution images.

**Figure 4 bioengineering-10-01258-f004:**
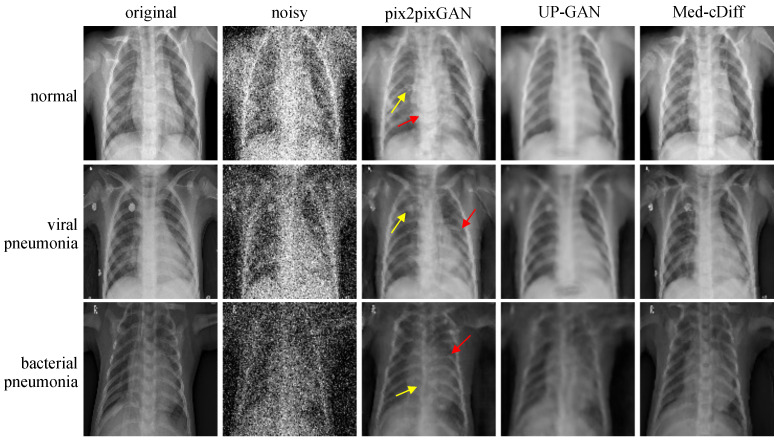
Qualitative comparison of Med-cDiff against other denoising methods. Arrows point to regions that pix2pixGAN cannot correctly generate.

**Figure 5 bioengineering-10-01258-f005:**
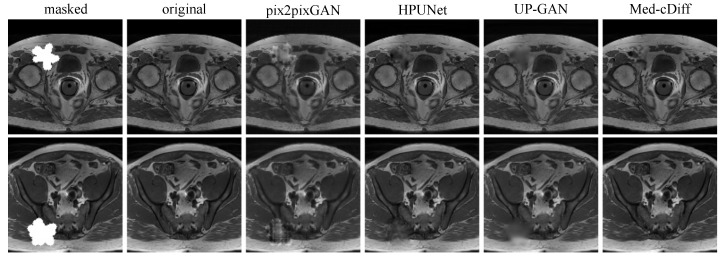
Qualitative comparison of Med-cDiff against other inpainting methods.

**Table 1 bioengineering-10-01258-t001:** Numerical comparison of Med-cDiff against other super-resolution methods.

Factor		LPIPS ($\times 10^{-4}$)↓	FID↓	acc.↑	PZ DSC (%)↑	TZ DSC (%)↑
$2\sqrt{2}$	bilinear	2787.847	**1.19**	6.75	**82.8**	87.7
	pix2pixGAN	**1.53**	1.20	12.72	81.5	87.2
	SRGAN	3.30	**1.19**	5.60	82.7	88.0
	Med-cDiff	2.74	**1.19**	**22.84**	81.7	**88.2**
4	bilinear	4339.392	1.20	4.51	78.2	84.2
	pix2pixGAN	**1.96**	1.22	11.31	78.3	86.1
	SRGAN	5.03	**1.19**	5.11	**80.2**	86.2
	Med-cDiff	4.62	**1.19**	**21.44**	77.8	**86.3**
$4\sqrt{2}$	bilinear	5773.238	1.21	3.28	68.9	75.9
	pix2pixGAN	**2.50**	1.22	12.68	69.2	81.1
	SRGAN	6.09	1.21	4.39	72.6	82.7
	Med-cDiff	5.09	**1.20**	**21.37**	**74.2**	**84.3**

**Table 2 bioengineering-10-01258-t002:** Quantitative comparison of Med-cDiff against other denoising methods.

	LPIPS ($\times 10^{-4}$)↓	FID↓	Classification Accuracy (%)↑
original image	-	-	70.7
noisy image	17.52	1.35	63.6
pix2pixGAN	1.77	1.32	65.1
UP-GAN	3.36	1.33	62.8
Med-cDiff	**1.19**	**1.30**	**65.8**

**Table 3 bioengineering-10-01258-t003:** Quantitative comparison of Med-cDiff against other inpainting methods.

	LPIPS ($\times 10^{-6}$)↓	FID↓	2AFC Accuracy (%)↓
pix2pixGAN	7.62	1.010	98.0
HPUNet	5.39	0.995	95.0
UP-GAN	3.17	0.897	94.5
Med-cDiff	**2.96**	**0.582**	**64.0**

## Data Availability

The raw data supporting the conclusions of this study will be made available by the authors in accordance with UCLA’s institutional management and sharing policy.

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
