# Peer review of "Med-cDiff: Conditional Medical Image Generation with Diffusion Models"

_bioengineering, 2023, doi:10.3390/bioengineering10111258_

Round 1

Reviewer 1 Report

Comments and Suggestions for Authors

This manuscript introduces a novel approach called Med-cDiff, a model applying conditional Denoising Diffusion Probabilistic Models for medical image generation tasks. The model presents nice performance in tasks such as super-resolution, denoising, and inpainting, outperforming certain existing generative models in certain metrics. 

In the manuscript, the author claims that certain state-of-art models are tailored to particular applications or imaging modalities. While you have highlighted that the propose model, Med-cDiff achieves state-of-the-art performance on various tasks, it would be beneficial to discuss further on the model's generalizability. For example, how does Med-cDiff overcome the limitations you pointed out in the previously discussed models?

Comments on the Quality of English Language

There appears to be some inconsistency in terminology throughout the manuscript. For example, terms such as "image-to-image translation" and "conditional image generation" are used interchangeably. Similarly, "state-of-the-art" and its abbreviation "SOTA" are both used, but "SOTA" is never defined, which may lead to confusion for some readers. 

Reviewer 2 Report

Comments and Suggestions for Authors

The paper is written in very good English.

The paper sufficiently cites relevant work in the medical imaging domain as well as in AI-based computer vision (Diffusion Models).

The method is fully formulated using equations in Section 3. Insights on their meaning are provided. 

The method is comparatively evaluated against the state-of-the-art on publicly available datasets.

In short, I think that the article is excellent.

I think however that the paper deserves some discussion, particularly on the following experimental condition. My understanding is that Med-cDiff trains a separate diffusion model for each dataset, in Section 4.1. Could there be a possibility of learning the individualities of the particular datasets? E.g. due to the way of image acquisition or the use of a particular modality? If this is not so, is it because the used datasets have great variability in the images they include? Conversely, if this is so, is there any way of excluding this possibility by testing on other data? In general, if somebody adopts Med-cDiff how rich should the training data be to avoid image generation errors/artefacts that could potentially affect diagnoses?
